

# Invasive crayfish does not influence spawning microhabitat selection of brown frogs

Samuele Romagnoli[1], Gentile Francesco Ficetola[1,2] and Raoul Manenti[1]

[1] Department of Environmental Science and Policy, University of Milan, Milan, Italy
[2] Laboratoire d'Ecologie Alpine (LECA), Université Grenoble-Alpes, Grenoble, France

Corresponding author
Samuele Romagnoli,
samuele.romagnoli@studenti.unimi.it

## ABSTRACT

Microhabitat selection is a key component of amphibian breeding biology and can be modulated in response to the features of breeding sites and the presence of predators. Despite invasive alien species being among the major threats to amphibians, there is limited information on the role of invasive species in shaping amphibians' breeding microhabitat choice. The invasive red swamp crayfish (*Procambarus clarkii*) is a major predator of amphibians' larvae, including those of the brown frogs *Rana dalmatina* and *Rana latastei*. Although qualitative information about the spawning site preferences and breeding microhabitat choice of brown frogs is available in the literature, only a few studies performed quantitative analyses, and the relationship between microhabitat choice and the presence of alien predators has not been investigated yet. The aims of this study were: (1) to characterize the microhabitats selected for clutch deposition by *R. dalmatina* and *R. latastei* and (2) to test if the position and the aggregation of egg clutches differ in sites invaded or not invaded by *P. clarkii*. During spring 2017, we surveyed multiple times 15 breeding sites of both brown frogs in Northern Italy; in each site we assessed the features of the microhabitat where each egg clutch was laid, considering its position (distance from the shore, depth of the water column) and the degree of aggregation of clutches. In each site we also assessed the presence/absence of the invasive crayfish and the relative abundance in the breeding period. We detected egg clutches in all sites; the crayfish occurred in eight ponds. Our results showed substantial differences between the spawning microhabitat features of the two brown frogs: *Rana latastei* clutches showed a higher degree of aggregation and were associated with deeper areas of the ponds , while *Rana dalmatina* deposited more spaced out clutches in areas of the ponds that were less deep. For both species, spawning microhabitat features were not significantly different between sites with and without *P. clarkii*. Although we did not detect behavioural responses to *P. clarkii* in the choice of spawning microhabitat , additional studies are required to assess whether these frogs modulate other behavioural traits (e.g. during larval development) in response to the invasive predator.

## INTRODUCTION

Amphibians are among the taxa with the highest ratio of threatened and declining species, and their global decline has been the focus of many studies to quantify and understand

the causes of this phenomenon (*Ficetola, 2015*; *Scheele et al., 2019*). The increase of trade and tourism all over the world, which directly or indirectly facilitates the spread of alien animals and plants, is one of the strongest threats to native biodiversity (*Davis, 2003*), and amphibians are particularly sensitive to the impact of alien species (*Bellard, Cassey & Blackburn, 2016*). Invasive alien species (IAS) can have multiple impacts on native amphibians. Predatory IAS can feed on both larvae and adults of many amphibian species, spread diseases, and limit the trophic resources available (*Fisher, Garner & Walker, 2009*; *Hettyey et al., 2016*; *Kats & Ferrer, 2003*), potentially leading to detrimental effects on the whole ecosystem of the invaded sites (*Jackson et al., 2016*).

Adult amphibians are often able to detect the presence of predators and modulate their breeding activity to limit predation on their offspring. On the one hand, parents may actively select breeding sites with few predators, and this has been shown to increase offspring fitness (*Resetarits, 2005*; *Sadeh, Mangel & Blaustein, 2009*; *Segev et al., 2011*; *Winandy, Darnet & Denoël, 2015*). The selection of breeding sites with few predators can also frequent in amphibians that stay in water for very short periods, and that can detect predators through indirect cues (e.g., chemical cues) (*Resetarits, 2005*). On the other hand, breeding sites without predators are not always available, and similarity of habitat preferences between amphibians and their predators may force the former to breed in sites with predators. However, wetlands can be very heterogeneous environments with a high number of microhabitats. Within a given breeding wetland the density of predators and predation risk can vary across microhabitats, thus females can select specific sites to increase the survival of tadpoles (*Ficetola, Valota & De Bernardi, 2006*). However, up to now very few studies have investigated whether amphibians change their patterns of microhabitat selection in response to invasive predators.

The red swamp crayfish *Procambarus clarkii* is among the IAS with the strongest impact both in Europe and on a global scale (*Nentwig et al., 2018*). This crayfish is a generalist feeder (*Alcorlo, Geiger & Otero, 2004*; *Whitledge & Rabeni, 1997*), and its global spread affects a growing number of freshwater communities worldwide (*Cruz et al., 2008*; *Ficetola et al., 2011b*; *Manenti et al., 2019b*; *Ramamonjisoa, Rakotonoely & Natuhara, 2018*; *Vilà et al., 2010*). *P. clarkii* preys on tadpoles of several amphibian species (*Cruz et al., 2008*); in northern Italy crayfish predation is a major cause of local extinctions and tadpole abundances reduction (*Ficetola et al., 2011b*). The strong predation of *P. clarkii* on frog tadpoles may drive rapid behavioural or evolutionary responses in invaded communities (*Nunes et al., 2014a*; *Nunes et al., 2014b*); on the other hand, when there is a lack of anti-predatory response to this invasive crayfish, there is a high risk of extinction of amphibian populations (*Nunes et al., 2013*). The negative relationship between *P. clarkii* and tadpole abundance is particularly evident for some brown frog species, such as the agile frog (*Rana dalmatina)* and the Italian agile frog (*Rana latastei*) (*Ficetola et al., 2011b*). Although these two species of brown frogs often breed in sites invaded by *P. clarkii*, a study performed a few years after the invasion showed that very few tadpoles reach metamorphosis in wetlands with high crayfish density (*Ficetola et al., 2011b*). *Rana dalmatina* and *R. latastei* are species of conservation concern, and their tadpoles can be the most abundant vertebrates in small

wetlands, thus they likely allow a substantial exchange of biomass from woody parches to the ponds in which they breed and vice-versa (*Barzaghi et al., 2017*; *Gibbons et al., 2006*).

To assess whether *R. dalmatina* and *R. latastei* can modulate microhabitat selection in presence of invasive predators, we tested three hypotheses.

(1) Shelter hypothesis. Along the shore, riparian semi-aquatic vegetation, submerged branches and hiding elements are more abundant and can offer shelter from aquatic predators (*Dodd, 2010*; *Ficetola, Valota & De Bernardi, 2006*; *Manenti et al., 2017*). Therefore, if the shelter hypothesis is correct, we expect that in invaded ponds, frogs lay clutches closer to the pond edge.

(2) Deepness hypothesis. The crayfish is rarely active in the water column, thus laying clutches in deep water can reduce predation rate on eggs and tadpoles (*Cruz & Rebelo, 2005*). Therefore, this hypothesis predicts that, in invaded sites, frogs lay clutches in deeper water.

(3) Schooling hypothesis. When tadpoles hatch, they show high local density during their most vulnerable stage; thus, by laying clutches nearby other clutches, frogs can form large assemblages of clutches and tadpoles. This grouping strategy can provide advantages under high predation risk (e.g., confusion, enhanced group vigilance, diluted predation risk (*Lima & Dill, 1990*; *Nicieza, 1999*)). If the schooling hypothesis is right, we expect to find less distance between clutches in invaded wetlands.

To achieve our aims we first characterized the features of the microhabitats selected by the two frogs; the spawning habits and the microhabitat selection of these two species have been described in the literature, but quantitative analyses remain limited (*Ancona & Capietti, 1996*; *Ficetola, Valota & De Bernardi, 2006*). We then tested if there was significant variation in spawning features between invaded and non-invaded sites.

## MATERIAL AND METHODS

In spring 2017, we surveyed 15 breeding sites in northern Italy for which previous surveys confirmed the reproduction of at least one brown frog species (*R. latastei* or *R. dalmatina*). All the wetlands were in the basin of two tributaries of the Po river, Lambro and Adda rivers, north of Milan. Since the early 2000s, wetlands of the Po lowland have been invaded by *P. clarkii* (*Fea et al., 2006*) that was first detected in our study area around 2005 (*Manenti et al., 2014*). Since 2005, *P. clarkii* spread in the study area colonising approx. 65% of the permanent, large ponds that generally constitute the breeding sites of these frogs (*Manenti et al., 2014*; *Manenti et al., 2020*). The crayfish impact is heterogeneous across sub-populations (*Manenti et al., 2020*; *Siesa et al., 2011*). Previous studies have shown *P. clarkii* exerts a heavy predation pressure on the larvae of *R. latastei* and *R. dalmatina*, strongly reducing their abundance, and that the impact of crayfish predation is stronger than the effects of native predators such as dragonflies (*Ficetola et al., 2012*).

In both frog species, the deposition period begins in early spring and each female lays only one clutch. The detectability of clutches is high and the two species can be identified based on of their morphological characteristics (*Ambrogio & Mezzadri, 2018*). For each site, we performed two surveys at the peak of frog breeding activity (March), one during

daytime and one during the night. During daytime surveys, we assessed the features of the microhabitat of deposition of each egg clutch of *R. latastei* and *R. dalmatina*. For each clutch we measured three variables describing the position and the degree of aggregation of clutches: distance from the pond shore, interclutch distance (distance from the closest conspecific clutch), and depth of the water column. We measured the distance from the pond shore as the minimum distance between the clutch and the closest edge of the breeding site. In the study ponds, aquatic vegetation was nearly absent, while semi-aquatic vegetation and submerged branches were most abundant near the shoreline, thus distance from the pond shore is a good proxy of the availability of shelters for tadpoles. To assess interclutch distance, for each clutch we identified the closest conspecific clutch and measured the distance between them. When two clutches were in contact, the distance was recorded as zero. We measured depth of the water column as the total depth of the water column at the spawning point, also if the clutch was underwater (as often occurs for *R. latastei*). Moreover, we measured the maximum depth and the surface of each wetland to compare pond features among invaded and not invaded sites. The total number of clutches per site and surface was then used to calculate clutch density at each breeding site. A few clutches were clearly laid several days before sampling, and, especially for the agile frog, drift could have modified their position; for this reason, they were not considered for microhabitat measurements.

To verify the occurrence of the red swamp crayfish and to measure the relative abundance of crayfishes in ponds active during the breeding period, we performed one visual encounter survey during night-time to maximize the detection probability of the crayfish. Recent studies showed that during nocturnal surveys the per-visit detection probability is very high (>95%; *Manenti et al., 2019b*). Surveys were performed using night lamps along the whole perimeter of the ponds and lightening the inner sectors as much as possible.

Invasive species often have the strongest impacts on sites where they attain the highest abundance (*Leung et al., 2012*). In this case, just measuring the presence/absence of invasives can obscure patterns caused by variation of abundance. Therefore we estimated relative abundance of crayfishes across sites during breeding period using CPUE (Catch Per Unit Effort) index (*Zimmerman & Palo, 2011*) applying the following formula:

$$\text{CPUE} = \frac{N_{\text{crayfish}}}{m \times t \times N_{\text{obs}}}$$

where "$N_{\text{crayfish}}$" is the number of individuals observed, "$m$" is the distance travelled during the survey, "$t$" the time spent in the survey and "$N_{\text{obs}}$" the number of observers participating in the survey (*Anderson, Paszkowski & Hood, 2015*).

Total survey time was proportional to the surface of ponds and considered in the calculation of the CPUE index. Particular attention was paid to the occurrence of small young individuals. Additional surveys were performed in the same sites during the late spring–middle summer of 2017 and 2019 and confirmed the absence of alien crayfish detection in the sites considered as non-invaded during this study (*Manenti et al., 2019b*). For only one pond we don't have data of relative abundance of crayfish.

## Statistical analyses

Before running analyses, interclutch distance, clutch density, distance from the shore and CPUE index were log-transformed, while water depth was square-root transformed to improve normality. First, we used $t$-tests, assuming heterogeneous variance, to assess whether pond features (pond surface, maximum depth and clutch density) were significantly different between invaded and non-invaded ponds. Then, we used linear mixed models (LMMs) to assess whether the study species select different microhabitat features. We ran three separate LMMs with the different dependent variables (distance from pond edge, interclutch distance and water depth) and with species identity as the independent variable; the site was included as a random factor to take into account the non-independence of clutches within the same site.

Finally, we tested whether spawning site selection is different between invaded and non-invaded sites. Given that we generally found differences in microhabitat selection between the two frog species (see 'Results'), we analysed them separately. For each species, we used LMMs to assess whether distance from the shore, interclutch distance and water depth are significantly different between invaded and non-invaded sites. For the analysis of water depth, we included the maximum water depth in each pond as a covariate; for the analysis concerning distance from the shore and interclutch distance we included clutch density as a covariate. In some cases, the variance of dependent variables showed heteroscedasticity between groups (*R. dalmatina* vs. *R. latastei* clutches; invaded vs non-invaded sites; see Supplemental Information for details). Therefore, we compared LMMs assuming homogeneous variance with models assuming heterogeneous variance between invaded and non-invaded sites. Models were fit using the VarIdent argument of the lmer function in R (*Pinheiro & Bates, 2000*). We used a likelihood ratio test to assess if the model assuming heterogeneous variance performed significantly better than the one with homogeneous variance. The model with heterogeneous variance was then used since it provided a significantly better fit. We used a likelihood ratio test to evaluate whether the model with heterogeneous variance better fit the data. This analysis was also repeated for each frog species using crayfish relative abundance (CPUE) instead of occurrence/absence as an independent variable (we excluded one breeding site of *R. dalmatina* because we didn't have data of abundance of crayfish). This allowed us to test the robustness of our conclusion to variation of crayfish abundance across ponds.

Analyses were performed in environment R using the packages "lme4" (*Bates et al., 2015*) and "nlme" (*Pinheiro et al., 2016*; *R Development, 2010*).

## RESULTS

Overall, we measured spawning site features for 498 clutches of the two frog species (333 *R. dalmatina* and 165 *R. latastei* clutches) in the 15 surveyed sites. Ponds showed an average ($\pm$SD) surface of 94.6 $\pm$ 87.1 m², a maximum depth in average ($\pm$SD) of 48.9 $\pm$ 24.7 cm. We detected the red swamp crayfish in eight sites. Regarding *Rana dalmatina*, we detected clutches in 13 sites, 6 uninvaded and 7 invaded by crayfish, with a mean ($\pm$SD) number of clutches per site of 26 $\pm$ 7. For *Rana latastei* we detected clutches

in 8 waterbodies, 6 uninvaded and 2 invaded (average: 21 $\pm$ 8 clutches per site). Ponds invaded and non-invaded by *P. clarkii* showed similar environmental variables. There were no significant differences for pond surface ($t_{9.2} = 2.05$, $P = 0.07$), max depth ($t_{11} = -1.46$, $P = 0.17$), or for clutch density of the two frog species (*R. latastei*: $t_{6.20} = -0.75$, $P = 0.48$, *R. dalmatina*: $t_{12.4} = -1.05$, $P = 0.31$). The CPUE index (abundance) of *P. clarkii* was generally low with a mean of 0.02 $\pm$ 0.03 (*N* crayfish/meters * minutes * *N* observers) for *R. dalmatina* breeding sites and 0.008 $\pm$ 0.02 for *R. latastei* breeding sites.

### Spawning microhabitat differences between *R. dalmatina* and *R. latastei*

*Rana latastei* clutches were more grouped than the *R. dalmatina* ones. The average interclutch distance ($\pm$ SD) was 24.7 $\pm$ 34.1 cm for *R. latastei*, and 98.3 $\pm$ 122.3 cm for *R. dalmatina* ($F_{1,482} = 29.1$; $P < 0.001$; Fig. 1A). Furthermore, *R. latastei* clutches were laid in deeper sectors of ponds than *R. dalmatina* ($F_{1,482} = 5.33$; $P = 0.02$; Fig. 1B); the mean depth ($\pm$ SD) of water column was 22.8 $\pm$ 7.8 cm for *R. latastei* clutches while it was 18.5 $\pm$ 6.8 cm for *R. dalmatina* clutches. *R. latastei* also laid clutches more distant from the shore of the pond than *R. dalmatina* ($F_{1,482} = 4.16$; $P = 0.04$; Fig. 1C); the mean ($\pm$SD) distance between the shore and *R. latastei* egg-clutches was 164.6 $\pm$ 201.6 cm, while it was 131.4 $\pm$ 101.8 cm for *R. dalmatina* clutches.

### Clutch microhabitat features in invaded and non-invaded sites

For both frog species, the spawning microhabitat features were similar between invaded and non-invaded ponds (Table 1). We did not detect any significant relationship between crayfish occurrence and microhabitat neither regarding the distance between the clutches, nor the depth of the water column, nor the distance from the pond edge. Results were identical when we consider the variation of crayfish abundance, as we did not detect any significant relationship between crayfish CPUE and spawning microhabitat features (Table 2).

## DISCUSSION

The differences observed between *Rana latastei* and *R. dalmatina* highlight the importance of microhabitat selection for these frog species that adopt different strategies and select different spawning sites. Nevertheless, we did not detect any relationship between the spawning microhabitat choice and the occurrence of the alien predator *Procambarus clarkii*. Our results did not conform with the predictions of either the shelter, the deepness or the schooling hypothesis, suggesting that the choice of the spawning position by adults is not modulated to minimize predation risk of eggs and tadpoles by the alien crayfish.

There are many reported cases of dramatic effects of alien species introduction on native species, which range from behavioural shifts of native species (*Tiberti & Von Hardenberg, 2012*; *Winandy & Denoël, 2013*) to cascading effects on entire ecosystems, and can lead to the extirpation of entire species or communities (*Arribas, Díaz-Paniagua & Gomez-Mestre, 2014*; *Bonelli, Manenti & Scaccini, 2017*). Semi-aquatic organisms like amphibians can play important roles for nutrient exchanges between aquatic and terrestrial habitats

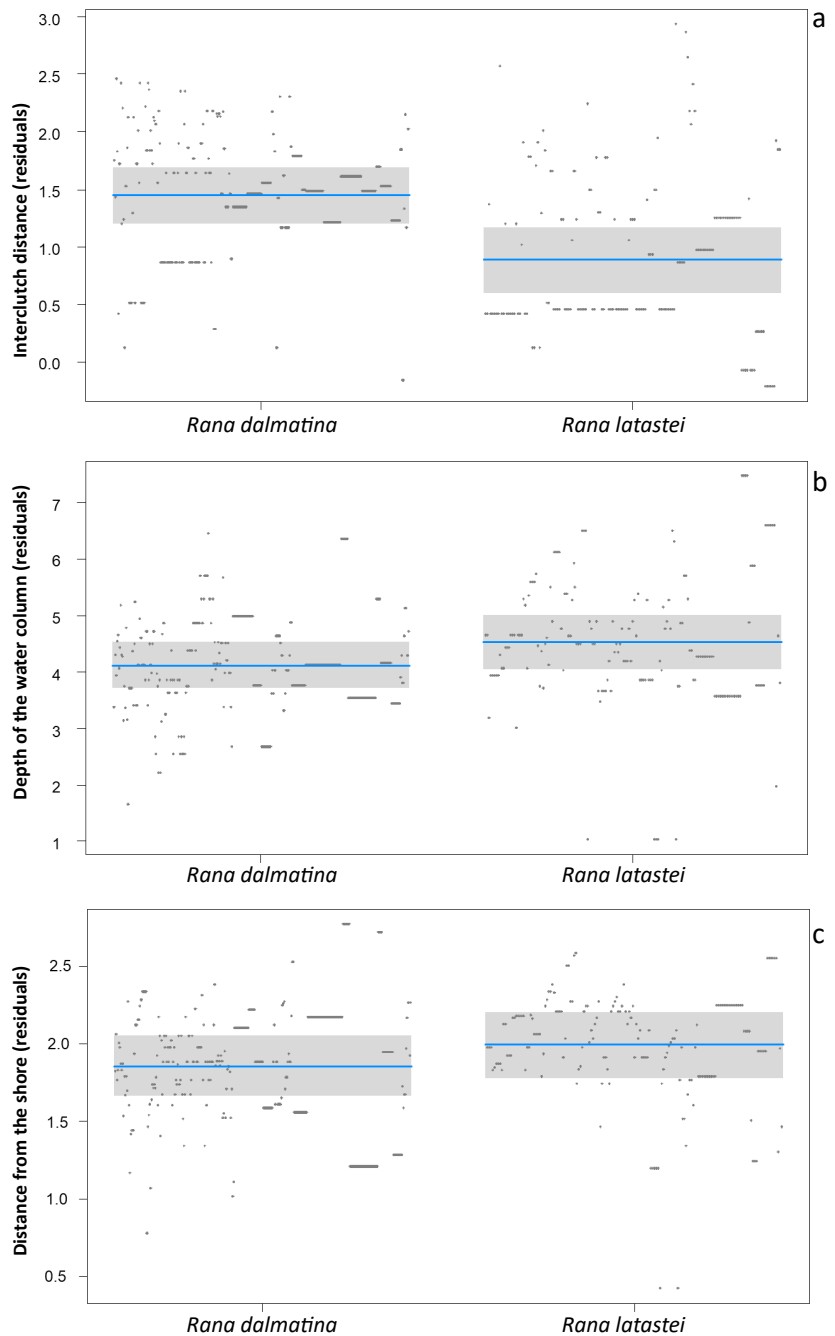

**Figure 1** **Differences between species.** Partial regression plots of models showing differences in clutch microhabitat features between *Rana latastei* and *R. dalmatina*. (A) Interclutch distance (residuals). (B) Depth of the water column (residuals). (C) Distance from the shore (residuals).

(*Barzaghi et al., 2017*; *Gibbons et al., 2006*), thus invasive crayfish is expected to deeply affect the biotic community of both lentic and lotic environments (*Ficetola et al., 2012*; *Ficetola et al., 2011b*; *Gherardi & Acquistapace, 2007*; *Manenti et al., 2019b*; *Shin-ichiro et al., 2009*). If the breeding microhabitat selected by adult amphibians is important for their

**Table 1  Clutch microhabitat features in invaded and non-invaded sites for each species. Mean values for each microhabitat features are expressed in cm ± SD.**

| Dependent | Independent | *Rana latastei* | | | | |
| --- | --- | --- | --- | --- | --- | --- |
| | | **Without crayfish Mean** | **With crayfish Mean** | **df** | **F** | **P** |
| Interclutch distance | | 10.0 ± 9.04 (cm) | 13.38 ± 36.15 (cm) | | | |
| | *P.clarkii* | | | 1, 5 | 0.23 | 0.65 |
| | Clutch density | 2.9 ± 0.2 | 0.4 ± 0.02 | 1, 5 | 0.43 | 0.54 |
| Water depth | | 21.7 ± 14.8 (cm) | 17.2 ± 9.2 (cm) | | | |
| | *P. clarkii* | | | 1, 5 | 0.06 | 0.81 |
| | Max. pond depth | 60 | 64.19 ± 1.9 | 1, 5 | 0.43 | 0.54 |
| Distance from the shore | | 95.1 ± 125.4 (cm) | 95.4 ± 128.7 (cm) | | | |
| | *P. clarkii* | | | 1, 5 | 0.01 | 0.91 |
| | Clutch density | 2.9 ± 0.2 | 0.4 ± 0.02 | 1, 5 | 0.77 | 0.42 |

| Dependent | Independent | *Rana dalmatina* | | | | |
| --- | --- | --- | --- | --- | --- | --- |
| | | **Without crayfish Mean** | **With crayfish Mean** | **df** | **F** | **P** |
| Interclutch distance | | 87.0 ± 62.3 (cm) | 26.0 ± 48.9 (cm) | | | |
| | *P.clarkii* | | | 1, 10 | 1.51 | 0.25 |
| | Clutch density | 0.7 ± 0.01 | 0.6 ± 0.06 | 1, 10 | 0.79 | 0.21 |
| Water depth | | 19.7 ± 7.8 (cm) | 20.9 ± 9.4 (cm) | | | |
| | *P. clarkii* | | | 1, 10 | 0.25 | 0.63 |
| | Max. pond depth | 61.6 ± 0.8 | 59.5 ± 2.1 | 1, 10 | 3.52 | 0.09 |
| Distance from the shore | | 168.7 ± 171.2 (cm) | 117.1 ± 189.5 (cm) | | | |
| | *P. clarkii* | | | 1, 10 | 0.25 | 0.63 |
| | Clutch density | 0.7 ± 0.01 | 0.6 ± 0.06 | 1, 10 | 1.71 | 0.22 |

breeding success, we expect a modulation of habitat choice in response to the presence of IAS. Evolutionary and plastic changes in invaded communities can arise at different levels and might be difficult to detect (*Nunes et al., 2014a*). In our study area, previous research detected brown frogs breeding in sites with red swamp crayfish (*Ficetola et al., 2011b*). Given the high tadpole mortality in invaded wetlands, a plastic selection of breeding habitats and microhabitats could limit predation on eggs or tadpoles. However, adult brown frogs have continued to breed for several years in these ponds, selecting the same microhabitat features as in non-invaded waterbodies. This suggests that, contrary to what happens in urodeles (*Cabrera-Guzman, Diaz-Paniagua & Gomez-Mestre, 2019*; *Winandy, Legrand & Denoël, 2017*), predation pressure of the alien crayfish does not lead to rapid shifts or responses in adult brown frogs spawning behaviour.

The ability of native prey to assess risk and adopt appropriate behavioural responses depends on different factors such as the experience accumulated during the lifespan, their learning ability and also their evolutionary history and ecology (*Kovacs et al., 2012*). Several factors can explain the lack of microhabitat shifts in brown frogs. First, contrary to urodeles that invest considerable time in courtship and eggs laying, brown frogs invest

**Table 2  Clutch microhabitat features and abundance of crayfish.** Clutch microhabitat features and effects of the relative abundance of crayfish for each species. Mean values for each microhabitat features are expressed in cm ± SD.

| Dependent | Independent | *Rana latastei* | | | | |
|---|---|---|---|---|---|---|
| | | Without crayfish Mean | With crayfish Mean | df | F | P |
| Interclutch distance | | 10.0 ± 9.04 (cm) | 13.38 ± 36.1 (cm) | | | |
| | CPUE | | | 1, 5 | 3.40 | 0.12 |
| | Clutch density | 2.9 ± 0.2 | 0.4 ± 0.02 | 1, 5 | 1.21 | 0.32 |
| Water depth | | 21.7 ± 14.8 (cm) | 17.2 ± 9.2 (cm) | | | |
| | CPUE | | | 1, 5 | 0.05 | 0.82 |
| | Max. pond depth | 60 | 64.19 ± 1.9 | 1, 5 | 0.18 | 0.68 |
| Distance from the shore | | 95.1 ± 128.7 (cm) | 95.4 ± 125.4 (cm) | | | |
| | CPUE | | | 1, 5 | 1.10 | 0.34 |
| | Clutch density | 2.9 ± 0.2 | 0.4 ± 0.02 | 1, 5 | 1.42 | 0.28 |

| Dependent | Independent | *Rana dalmatina* | | | | |
|---|---|---|---|---|---|---|
| | | Without crayfish Mean | With crayfish Mean | df | F | P |
| Interclutch distance | | 87.0 ± 62.3 (cm) | 22.5 ± 42.8 (cm) | | | |
| | CPUE | | | 1, 9 | 3.10 | 0.11 |
| | Clutch density | 0.7 ± 0.01 | 0.6 ± 0.06 | 1, 9 | 0.04 | 0.83 |
| Water depth | | 19.7 ± 7.8 (cm) | 21.3 ± 9.3 (cm) | | | |
| | CPUE | | | 1, 9 | 1.05 | 0.31 |
| | Max. pond depth | 61.6 ± 0.8 | 59.5 ± 2.1 | 1, 9 | 1.15 | 0.27 |
| Distance from the shore | | 168.7 ± 171.2 (cm) | 117.7 ± 191.5 (cm) | | | |
| | CPUE | | | 1, 9 | 0.77 | 0.40 |
| | Clutch density | 0.7 ± 0.01 | 0.6 ± 0.06 | 1, 9 | 2.28 | 0.16 |

relatively little time in egg-laying, and frogs remain in breeding wetlands from few hours to at maximum one day (*Ambrogio & Mezzadri, 2018*). Such a period can be too short to allow them to acquire enough experience on the risk determined by the red swamp crayfish occurrence. Nevertheless, it is important to remark that multiple studies evidenced that amphibians can detect predator chemical cues in water, and can modulate the breeding habitat selection even when they use breeding sites for few hours (*Resetarits, 2005*; *Sadeh, Mangel & Blaustein, 2009*). Second, brown frogs breed at the end of winter, when water temperatures are low. The activity of the red swamp crayfish is generally high during warm periods (*Holdich et al., 2009*); thus, the individuals may not be particularly active when adult frogs are in the water, with limited consumptive effects. The worst effect of the crayfish probably happens only later in the season when temperatures rise, and the crayfish mostly impacts the tadpoles. Third, it is possible that the time since the arrival of the red swamp crayfish was not enough for frog populations to develop appropriate antipredator adaptations. Recent studies show that even if the ability to recognise invasive predators may evolve quickly, agile frog populations can be vulnerable to alien fish due to their inability to recognize them as a threat (*Hettyey et al., 2016*).
The lack of response in our target species can thus be explained by naïveté towards a novel predator (*Sih et al., 2010*). The first detection of the red swamp crayfish in the study area dates back to 2005 (*Manenti et al., 2014*), indicating that frogs coexisted with the crayfish for approx. 3–4 generations (*Guarino et al., 2003*; *Racca, 2003*; *Weddeling et al., 2005*). Rapid local adaptations of brown frogs when selective pressure is strong are known (*Ficetola et al., 2011a*; *Skelly & Freidenburg, 2000*), still amphibians responses to invasive species are mostly known to occur at the tadpole stage (*Hettyey et al., 2016*; *Nunes et al., 2014b*; *Nunes et al., 2013*). Future studies are required to investigate if behavioural responses may occur in the larval stages of the study species. We should also underline that, especially for *R. latastei*, the number of uninvaded ponds was low, because the crayfish has already invaded most breeding sites of this frog. To confirm the generality of our results it would be thus interesting to increase the number of uninvaded breeding sites, for instance in regions where *P. clarkii* is less widespread.

## CONCLUSION

Our study characterized the spawning microhabitat of the agile frog and the Italian agile frog, considering also sites in which they are syntopic and quantifying differences in spawning microhabitat between these two species. Italian agile frog females lay eggs in significantly deeper areas of the ponds, and with a strongly aggregated pattern. The clustering of Italian agile frog clutches has been repeatedly described in the herpetological literature, and it is known that dozens of females can attach their egg-clutches to the same submerged woods (*Ambrogio & Mezzadri, 2018*; *Pozzi, 1980*). However, there are few data on the causes of microhabitat selection. First, differences in microhabitat could reduce the frequency of interactions with heterospecific males, which in turn can reduce the fertility of clutches (*Ficetola & Bernardi, 2005*; *Hettyey & Pearman, 2003*; *Hettyey et al., 2014*). Second, tadpoles could exploit different microhabitats within the wetlands, for instance because they have different thermal optima (*Balogová & Gvoždík, 2015*). Nevertheless, very limited information exists so far on differences in microhabitat use between tadpoles, and future studies are required to understand the factors allowing the syntopy between these frog species, and the relative effect of micro and macro-ecological determinants. A better knowledge of behaviour of these species is an important starting point for a better understanding of the strategies of these animals in response to biological invasions.

The invasive crayfish is widespread in the study area (*Manenti et al., 2020*; *Manenti et al., 2019b*) and is likely to attain high density, making eradication programs almost impossible. Under these circumstances, it is essential to identify the processes that can allow long-term persistence of native species, such as behavioural changes or the selection of specific microhabitats. However, such processes can be complex and can occur at multiple levels, and this can make their identification challenging. On the one hand, it will be important to integrate analysis performed on the microhabitat-scale with research performed on the landscape level, with long term analysis including the metapopulation-scale (*Manenti et al., 2020*). On the other hand, additional studies are required to assess whether native frogs can modulate other behavioural traits when interactions with the crayfish are more frequent, such as during larval development.

## ACKNOWLEDGEMENTS

We thank Pietro Leotta, Vito Leotta, F Maurizio, P Togni and S Fumagalli for different linguistic revisions of the manuscript. The comments of the editor Claire Paris and two anonymous reviewers greatly improved an early draft of the manuscript.

### Funding

The authors received no funding for this work.

### Competing Interests

The authors declare there are no competing interests.

### Author Contributions

- Samuele Romagnoli conceived and designed the experiments, performed the experiments, analyzed the data, prepared figures and/or tables, authored or reviewed drafts of the paper, and approved the final draft.
- Gentile Francesco Ficetola conceived and designed the experiments, analyzed the data, authored or reviewed drafts of the paper, and approved the final draft.
- Raoul Manenti conceived and designed the experiments, performed the experiments, authored or reviewed drafts of the paper, and approved the final draft.

### Data Availability

The raw measurements of clutch microhabitat features are available in the Supplemental Files.

### Supplemental Information

Supplemental information for this article can be found online at http://dx.doi.org/10.7717/peerj.8985#supplemental-information.

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
