# Peer review of "Invasive crayfish does not influence spawning microhabitat selection of brown frogs"

_PeerJ, doi:10.7717/peerj.8985_

## Round 0.1 · original submission · Major Revisions

Testing hypotheses related to changes in reproductive behavior in the presence of invasive species is rather novel and should have broader impact beyond amphibians. However, your findings have to be supporter not only by clear hypotheses (which is the case) but also appropriate experimental design and statistics related to sample size and measurements. In addition, using distance from the ponds' shore as proxy for vegetation/shelter for the egg clutches needs to be well justified. Please carefully address all concerns the reviewers put forward and make sure to follow their suggestions.

You may want to send the manuscript for professional English proof reading and editing before submission of the revised version.

Reviewer 1 ·

Basic reporting

The manuscript explores the possibility of adaptive egg-laying behavior displayed by two frog species in ponds invaded by the red swamp crayfish. Studies exploring the coexistence between native and invasive species are highly relevant in the context of global changes, especially applied to threatened groups like amphibians. I appreciated this manuscript which is clear and presents well-defined hypotheses. However, while sample sizes are appropriate for the statistical analyses applied, I have concerns with the biological representativity of the results, because of the low number of ponds that were sampled within each factor modality. Additionally, I suggest to reconsider the proxy used for the shelter hypothesis. The main text lacks details on sample sizes and on the global context regarding the invasion pressure. Please see my major comments in the next section.
The text must be edited by a native English speaker.

Experimental design

(1) 15 ponds were considered for this study (8 invaded and 7 uninvaded ponds), for a total of 498 clutches sampled in two frog species. While these sample sizes are sufficient for the statistical models applied, I am worried with their biological representativity, because of the low number of ponds sampled: one species was found in 13 ponds, and the other is found in 8 ponds, with N < 10 clutches found in 7 ponds. In the third analysis, models are conducted separately for each species to test for the effect of crayfish, and clutches belong to a maximum of 7 sites per modality (invaded or uninvaded) and as low as 2 sites for Rana latastei in uninvaded ponds. This limited representativity should be discussed in the manuscript.

(2) In the introduction, the authors hypothesize that the unavailability of predator-free habitats could force amphibians to breed in invaded ponds, and to adopt adaptive strategies to limit predation on their progeny. This is a sound starting point for the manuscript. However, information on the predation pressure or overall crayfish occurrence in the study area is lacking, while it would considerably enlighten the results of the study. Only half of the 15 sampled ponds were invaded, suggesting that habitats free of crayfish are still largely available.

(3) The hypotheses of adaptive behavior are well argumented and accompanied of clear predictions. However I have concerns about using distance from pond edge as a proxy for aquatic vegetation cover. Presence of aquatic vegetation notably depends on pond depth, water turbidity and bank slope. The supplementary tables indicate very shallow ponds with a maximum pond depth of 80 cm. Because the authors used visual surveys to detect crayfish, I also assume that water turbidity is low (but see my next comment). Then, there is no particular reason to assume that vegetation is restricted to pond edges, and I suggest the authors to use direct information on vegetation. Alternatively, the authors should better justify the use of this proxy and discuss this point. Additionally, it seems to me that the shelter and deepness hypotheses should not be mutually exclusive, which is somewhat the case in the present version.

(4) Attaining high probabilities of crayfish detection using visual encounter surveys is highly dependent on water turbidity, habitat complexity, i.e. site characteristics, as well as methodology (e.g. walking pace, counts…), so that more information on the protocol used to detect crayfish is needed. If available, the authors could reinforce the certainty of non-detections if crayfish also went undetected in previous surveys, especially using other techniques such as traps. Additionally, because the effect of crayfish on larval densities has been shown to vary with its density (e.g. Ficetola et al. 2011), the authors should consider using crayfish density instead of presence/absence data. Whether this option is not feasible because of sample sizes, this point should be discussed with point (1).

Litterature cited:
Ficetola, G., Siesa, M. E., Manenti, R., Bottoni, L., De Bernardi, F., & Padoa-Schioppa, E. (2011). Early assessment of the impact of alien species: differential consequences of an invasive crayfish on adult and larval amphibians. Diversity and Distributions, 17(6), 1141–1151.

Validity of the findings

(5) More information on sample sizes should be directly available to help the reader judge the reliability of the study: number of ponds studied per species, and for each species, the number of invaded and uninvaded sites, directly in the text or in Table 2; mean +/- SD or range of clutches found per site per species. This information can currently only be calculated from the supplementary tables. Means and standard errors of the environmental variables can be added to help the reader visualize the study systems.

(6) The results on spawning microhabitat differences between species are means of means per site, while Figure 1 looks based on raw data. Please homogenize. Please add standard deviations to all the given means to help the reader judge the significance of the results.

(7) Line 237 “The invasive crayfish has attained very high density”. Vague sentence, please justify with data or reference.

(8) Table 1 could be placed in supplementary material as it stands for a methodological point.

(9) Please check values in the supplementary tables: there is a 0.9 maximum pond depth, and pond width is often larger than pond length. Add a legend to these tables that explains variables, units, and acronyms.

(10) Please cite the publications associated to the R packages used, which can be found using the citation(“pckg_name”) function in R.

(11) Please check the English language and typos: lines 63, 72, 81, 136-137, 154, 197-198, 202, 215-216, 226, 235-236... and column names in supplementary tables.

Reviewer 2 ·

Basic reporting

This paper by Romagnoli and colleagues reports on a study investigating the microhabitat preferences during egg-deposition in two anuran species, Rana dalmatina and R. latastei, and assessing whether these preferences are influenced by the presence/absence of Procambarus clarkii, an invasive alien species that is a voracious predator of anuran larvae. Because studies on such predator-mediated changes in habitat- and microhabitat use during egg-laying are rare in amphibians, and the spread of P. clarkii is becoming a major problem of conservation biology, the study is novel and interesting. The study design is appropriate, the introduction sets the scene well, and results underpin the interpretations. However, there are a few issues that need to be addressed.

First of all, the English needs to be corrected. I provided suggestions in this respect (please see attachment), but I am not a native English speaker, so that further language editing would certainly enhance the legibility of the paper.

A few important papers that should have been considered thoroughly are only mentioned tangentially, or are not referred to at all. For example, the papers Nunes et al. 2014 Ecology, Nunes et al. 2014 Ecology and Evolution, Nunes et al. 2013 Oecologia, Hettyey et al. 2016 Ecology should be cited and discussed both in the introduction and the discussion.

Experimental design

Lines 75-77: The authors state that according to the shelter hypothesis, because vegetation is usually more dense along the shores of ponds, shelters may be more abundant close to the shore, so that the expectation is that frogs would deposit their eggs close to the shore when P. clarkii is present in order to provide protection to their offspring. It is unclear, however, why the authors used the proxy of distance from the shore instead of directly measuring vegetation density around clutches. That would have been clearly preferable.

Lines 81-82: Regarding the schooling hypothesis: I find it difficult to believe that tadpoles' limited home ranges lead to tadpole aggregations when eggs are laid in close proximity. Tadpoles often can be seen at places where no clutches had been laid before within tens of meters. This statement at least needs some good citations. Or it would need to be changed in the sense that when tadpoles hatch and are present in high density locally during their most vulnerable stage (early post-hatching), their survival probabilities may be enhanced in the ways spelled out in the next sentence.

Lines 111-113: It is mentioned that maximum water depth was measured and clutch density was calculated, but it is unclear, why. Please explain here briefly.

Validity of the findings

Lines 111-113: It is mentioned that maximum water depth was measured and clutch density was calculated, but it is unclear, why. Please explain here briefly.

Line 161: One of the results is that R. latastei laid clutches into deeper water than R. dalmatina. However, Fig. 1b suggests the opposite. Please double-check and clarify.

Line 165: As before, the figure contradicts the result. Please double-check and clarify.

Line 193: The end of the discussion's first paragraph is left in empty space: a conclusion is missing.

Lines 204-208: Female frogs invest a lot of energy into the eggs, and males spend a lot on calling, fighting, maintaining a territory, etc. Also, it is the time factor that may be important here (enough vs. not enough time to reliably assess the presence and abundance of predators), so that the reference to relatively little investment in energy should be deleted.

Lines 211-212: This idea needs to be elaborated a bit more. It may not be imperative to adjust microhabitat preferences during egg-laying because of low predator activity during that cool period, while the devastating effect of the crayfish on theses amphibians may manifest later on during the tadpole stage, when water temperatures rise.

Lines 216-217: Awkward wording. Frogs (or any other organisms) do not evolve something. Evolution is not an active process in this sense. Evolution is not performed, it just happens.

For further minor comments, please see the attached file.

Additional comments

Table 1 presents data of marginal (only technical) importance, so that it should be moved into an electronic appendix.

Table 2: Dependents and independents are messed up, so that this table is difficult to read. Please re-arrange.

Annotated reviews are not available for download in order to protect the identity of reviewers who chose to remain anonymous.

---

## Round 0.2 · Major Revisions

· Appeal

Appeal


· · Academic Editor

Reject

I have to agree with the evaluation of Reviewer 1 that the experimental design and leading hypotheses are not in line. I am confident that the detailed comments of the reviewer will help in better framing the scope of your study

Reviewer 1 ·

Basic reporting

Line 54. The term “niche overlap” is confusing since it could refer to many dimensions: thermal niche, trophic niche… it seems to me that you specifically refer to habitat preferences.
Line 66. “The strong selection by P. clarkii on frog tadpoles”. Do you have any reference indicating that P. clarkii selects tadpoles over another resource? Or do you mean natural selection of tadpole behaviour? Or do you simply mean “the strong predation by P. clarkii”? Be more specific.
Lines 79-83. Your detailed answer raised an important complementary point: if aquatic vegetation was absent from your ponds, then shores were the only refuges in the pond. This should appear in the text to build your hypothesis.
Lines 104-107. Based on the surveys mentioned, can you provide precise estimates on crayfish occurrence? (e.g. number of invaded ponds and total number of ponds). It would be useful to provide it here to judge of the availability of uninvaded habitats.
Line 170. Unit error?
There are again careless mistakes throughout the manuscript: typos in the text and in table column names, double parentheses in citations, wrong units throughout the manuscript and table legend.

Experimental design

In the light of your modifications of the manuscript and your point-by-point response, it seems to me that the working hypothesis of the manuscript is highly questionable.
You specified in your answer that crayfish activity (and catchability) was so reduced during frog spawning that you cannot even obtain estimates of crayfish abundances. You also stated that frog breeders stay in ponds a maximum of one day. Then how could frog breeders detect crayfish, let alone change their spawning strategy in response?
These elements should rather be a starting point to establish the working hypothesis that these frogs cannot adapt their spawning strategy to this invasive species. Note that this has already been suggested in Ficetola et al. 2011 (Divers and Distrib).
There is also a confusion between adult strategies and impact/adaptive response of tadpoles. The latter is described throughout the manuscript, but it seems to be out of the scope of the manuscript – again, frog breeders do not stay in ponds after spawning.

Validity of the findings

Lines 106-108. Are you suggesting that invaded and uninvaded ponds have different hydroperiods? This has major implications for your results.
Line 134. Pond surfaces vary from 8 to 283 m². (Note that the unit indicated in legend is cm²). Did you adapt your survey time to pond surface to homogeneize detection probabilities?
Line 180-187. Variability is huge for interclutch distances and distance to pond edges! Indeed, average measures are significantly different, but conclusions must be more elaborated than “clear differences between R. l. and R. d.”.
Line 184. What is the biological implication of such a small depth difference, even significant?

Additional comments

Investigating whether native species can adopt strategies to avoid invasive species is an important and promising field research. I encourage the authors to identify plausible mechanisms for the adaptation of particular species before testing different hypotheses.

Reviewer 2 ·

Basic reporting

No comment.

Experimental design

No comment.

Validity of the findings

No comment.

Additional comments

The authors addressed my concerns adequately, I have no further comments to make.

---

## Round 0.3 · Minor Revisions

The study indicates significant differences between the spawning microhabitat features of the two brown frogs investigated, and no effect of the presence of predator on the spawning location of two species of frogs in ponds. This results should be clearly stated an indicated in the title, currently "Presence of invasive crayfish and spawning site selection in brown frogs" - suggested title: "Invasive crayfish does not influence spawning site selection of brown frogs". The manuscript sill requires minor revisions, mostly editorial. I recommend to send the manuscript to a professional editing service before resubmitting.

---

## Round 0.4 · accepted · Accept

Thank you for your patience on the decision.